# Transcriptional Regulation of Siglec-15 by ETS-1 and ETS-2 in Hepatocellular Carcinoma Cells

**DOI:** 10.3390/ijms24010792

**Published:** 2023-01-02

**Authors:** Kaiqin Sheng, Yuecheng Wu, Hanbin Lin, Menghan Fang, Chaorong Xue, Xu Lin, Xinjian Lin

**Affiliations:** 1Key Laboratory of Gastrointestinal Cancer (Fujian Medical University), Ministry of Education, Fuzhou 350122, China; 2Fujian Key Laboratory of Tumor Microbiology, Department of Medical Microbiology, Fujian Medical University, Fuzhou 350122, China

**Keywords:** Siglec-15, TGF-β1, ETS-1, ETS-2, HCC, transcriptional regulation

## Abstract

Sialic acid-binding immunoglobulin-like lectin 15 (Siglec-15) has been identified as a crucial immune suppressor in human cancers, comparable to programmed cell death 1 ligand (PD-L1). However, the regulatory mechanisms underlying its transcriptional upregulation in human cancers remain largely unknown. Here, we show that the transcription factors ETS-1 and ETS-2 bound to the Siglec-15 promoter to enhance transcription and expression of Siglec-15 in hepatocellular carcinoma (HCC) cells and that transforming growth factor β-1 (TGF-β1) upregulated the expression of ETS-1 and ETS-2 and facilitated the binding of ETS-1 and ETS-2 to the Siglec-15 promoter. We further demonstrate that TGF-β1 activated the Ras/C-Raf/MEK/ERK1/2 signaling pathway, leading to phosphorylation of ETS-1 and ETS-2, which consequently upregulates the transcription and expression of Siglec-15. Our study defines a detailed molecular profile of how Siglec-15 is transcriptionally regulated which may offer significant opportunity for therapeutic intervention on HCC immunotherapy.

## 1. Introduction

Hepatocellular carcinoma (HCC) is the second leading cause of cancer-related mortality worldwide. In the recent years, systemic therapy for advanced HCC has improved patients’ prognosis due to deeper understanding of HCC molecular mechanisms, leading to the development of more effective therapeutic options including tyrosine kinase inhibitors and immunotherapy. During the last decade, special attention has been converged on immune checkpoint inhibitors (ICIs), as exemplified by anti-PD-1/PD-L1 (programmed cell death-1/programmed cell death ligand 1), that have revolutionized the way to manage advanced HCC. The approach of blocking PD-1/PD-L1 has highlighted the importance of tumor immune-evasion mechanisms that could be targeted to restore a robust T-cell response at the tumor site [1,2]. However, despite the great success of this unique approach, most patients develop primary or acquired resistance, calling for other immune modulatory pathways that may act cooperatively or alternatively to PD-1/PD-L1 blockage to augment the success of cancer immunotherapy.

Siglecs are a family of immunoglobulin-like lectins that can specifically bind to glycans and possess immunosuppressive properties on immune cells [3]. Recently, an increasing body of evidence has pinpointed a crucial role of Siglec members in tumor immunosuppression [4]. Among the Siglec family members, Siglec-15 is of particular interest in cancer immunity because of its substantial expression on the surface of macrophages and cancer cells, which is mutually exclusive to that of PD-L1, implicating that it may function as a crucial immune evasion mechanism in patients with negative PD-L1 expression. In fact, targeting Siglec-15 appears to be an alternative but effective therapy for patients who do not respond to the treatment of PD-1/PD-L1 antibodies [5,6,7]. A humanized monoclonal antibody against Siglec-15, named NC318, has been evaluated in phase I clinical trials for patients with advanced or metastatic solid tumors. However, such a trial was conducted without any biomarker and the molecular mechanisms underlying Siglec-15 expression and regulation for the therapeutic exploitation remain largely elusive.

The ETS family of transcription factors have been shown to play an important role in development and progression of various cancers by regulating the expression of genes involved in signaling pathways, cell proliferation, differentiation, apoptosis, invasion and metastasis, etc. [8]. ETS-1 and ETS-2 are the two closely related ETS family members with a highly conserved DNA-binding domain, i.e., ETS domain, which recognizes sequences having a GGAA/T core motif of DNA [9]. Basically, all ETS family members could bind to DNA sequences containing this core motif, and a majority of cell types express multiple ETS members [8]. Actually, many promoters contain ETS/AP-1 composite binding sites that are capable of mediating responsiveness of promoters to the Ras/C-Raf/MEK/ERK1/2 signaling pathways in the context of ETS-1 and other ETS proteins [9]. Furthermore, some members of the ETS family could serve as mediators of the MAPK/ERK pathway, which modulate expression of certain genes via phosphorylation. In particular, it has been identified that a single MAPK phosphorylation site is located near the pointed ETS-2 domain.

In this study, we aimed to investigate the molecular mechanisms of how Siglec-15 was transcriptionally regulated in HCC. We found that Siglec-15 was upregulated in HCC tissues and hepatocarcinoma cells. Mechanistically, increased expression of Siglec-15 in HCC cells was mediated through the transcription factors ETS-1 and ETS-2 that bound to the specific region of Siglec-15 promoter. Moreover, TGF-β1 stimulation enhanced the transcription and expression of Siglec-15 via activation of Ras/C-Raf/MEK/ERK1/2 pathway that phosphorylated and activated ETS-1 and ETS-2 to further enhance Siglec-15 expression.

## 2. Results

### 2.1. Siglec-15 Is Highly Expressed in the HCC Tissues and Cells

To assess the clinical significance of Siglec-15 expression, we first analyzed the Siglec-15 expression level using The Cancer Genome Atlas (TCGA) data of 369 HCC patients and found that Siglec-15 expression was significantly upregulated in the HCC tissues compared with their adjacent normal tissues (Figure 1A). We then examined the expression level of Siglec-15 in the two HCC cell lines, HepG2 and Huh7. As shown in Figure 1B,C, both mRNA and protein expression levels of Siglec-15 were upregulated in the HCC cells as compared to LO2 cells, the normal embryonic liver cells. These results imply that Siglec-15 is frequently upregulated in HCC and may highlight the importance of Siglec-15 expression for HCC immune modulation.

### 2.2. Siglec-15 Gene Promoter Activity in HCC Cells and Putative Transcription Factor Binding Sites

To examine the transcriptional activities of the Siglec-15 promoter, the full length (nucleotides −1800 to +55 nt) and a series of 5′-deletion constructs of the Siglec-15 promoter were transfected transiently into the human hepatocarcinoma cell line HepG2. As shown in Figure 2A, cells demonstrated the highest relative luciferase activities when transfected with pGL4.10-453 whereas the luciferase activity was dramatically reduced when the cells were transfected with pGL4.10-231, suggesting that this short region that contained nucleotides −453 to −232 nt was crucial for retaining the activity of the Siglec-15 promoter. When the JASPAR, Gene-regulation and PROMO programs were utilized to search for putative transcription factor binding sites, six potential ETS binding sites (i.e., ETS-1 and ETS-2) were identified in the −453 to −232 nt region of the promoter (Figure 2B). We next examined the effects of ETS-1 and ETS-2 expression on Siglec-15 promoter activity by overexpressing or knocking down ETS-1 or ETS-2 in HepG2 and Huh7 cells and the efficiency of overexpression or knockdown was confirmed by Western blot analysis (Figure 2C,D). We found that overexpression of ETS-1 or ETS-2 significantly increased Siglec-15 promoter activity (Figure 2E) whereas knockdown of ETS-1 or ETS-2 produced the opposite effect (Figure 2F). Since there were six putative ETS-binding sites (EBSs) within the Siglec-15 promoter (Figure 2B), we went further to determine whether the transactivation of the Siglec-15 promoter by ETS-1 and ETS-2 depended on these binding motifs by mutating the EBSs and measuring the Siglec-15 promoter activities. As shown in Figure 2G, mutations in each of EBSs resulted in a decrease of the promoter activity by approximately two-fold compared with the non-mutated control construct (pGL4.10-453-Luc). Of note, mutation of one single site caused only moderate loss in the luciferase activity since the other five binding sites may still function to drive the luciferase transcription. To further confirm the binding of ETS-1 and ETS-2 specifically to each of these EBS regions, EMSA analysis was performed using the biotin-labeled EBS probes containing the respective ETS-1 and ETS-2 recognition sequences and nuclear protein extracts from the HepG2 cells. As shown in Figure 2H, the retarded signals, assuming the formation of DNA-transcription factor complex, appeared dependent on the presence of the nuclear extracts (Lane 2), and the competition study demonstrated that a 100-fold molar excess of cold wild-type EBS probes but not the mutated EBS probes decreased the band intensity (Lane 3 and 4). Moreover, addition of antibodies against ETS-1 or ETS-2 could further shift up the dsDNA-protein complex (Lane 5 or 6) formed when the biotin-labeled EBS reacted with the nuclear protein extract from HepG2 cells. These results suggest that the ETS-1 and ETS-2 transcription factors might bind specifically to the EBS regions of the Siglec-15 promoter.

### 2.3. ETS-1 and ETS-2 Enhance Transcription of Siglec-15 in HCC Cells

To test whether the individual ETS-1 or ETS-2 was indeed essential for Siglec-15 transcription, both mRNA and protein expression levels of Siglec-15 were measured by quantitative real-time PCR and Western blot analysis in HepG2 or Huh7 cells with ETS-1 or ETS-2 overexpressed or knocked down. As anticipated, overexpression of ETS-1 or ETS-2 significantly increased transcription of Siglec-15 at both mRNA and protein levels in HepG2 and Huh7 cells (Figure 3A,C). In sharp contrast, when endogenous expression of ETS-1 or ETS-2 was knocked down by siRNA, Siglec-15 mRNA and protein levels were markedly reduced (Figure 3B,D). Notably, the specificity and efficiency of ETS-1 and ETS-2 overexpression or knockdown per se at both mRNA and protein levels were confirmed by qRT-PCR (Appendix A) and Western blot analysis (Appendix A). This observation is consistent with the concept that ETS-1 or ETS-2 could regulate Siglec-15 transcription through binding to the respective EBS in the Siglec-15 promoter.

### 2.4. TGF-β1 Stimulates Siglec-15 Expression in HCC Cells

TGF-β1 has been reported to play an essential role in immunosuppression and associates with poor prognosis in cancer patients [10]. To investigate whether TGF-β1 could as well mediate Siglec-15 expression in HCC cells, the expression levels of Siglec-15 in both HepG2 and Huh7 cells were examined by Western blot and flow cytometric analysis. As shown in Figure 4A,B, treatment of the cells with TGF-β1 significantly increased Siglec-15 expression. Quantitative RT-PCR analysis exhibited a significant increase of Siglec-15 expression in the response to TGF-β1 treatment (Figure 4C). Consistently, with a Siglec-15 promoter-luciferase reporter assay, we found that TGF-β1 promoted luciferase activity in the HCC cells (Figure 4D). Of note, a selective TGF-β1 receptor inhibitor SB431542 could reverse the effect of TGF-β1 on Siglec-15 expression. These data indicate that TGF-β1 positively regulated Siglec-15 expression at transcript levels in HCC cells.

### 2.5. ETS-1/ETS-2 Is Indispensable for TGF-β1-Induced Transcriptional Activation of Siglec-15

Given that TGF-β1 enhances transcriptional activities of Siglec-15, we attempted to elucidate molecular mechanisms underlying the regulation of Siglec-15 expression by TGF-β1. To this end, a full length and a series of 5′-deletion constructs of the Siglec-15 promoter fused to the luciferase reporter gene were created and transfected into HepG2 cells then treated with TGF-β1. As shown in Figure 5A, TGF-β1 significantly induced the luciferase activities in the cells transfected with pGL4.10-1800, pGL4.10-1540, pGL4.10-1270, pGL4.10-942, pGL4.10-735, pGL4.10-453, pGL4.10-231. In contrast, there was no effect of TGF-β1 on luciferase activity in the HepG2 cells transfected with pGL4.10-21, suggesting that the region of −231 to −22 nt was critical for transcriptional activation of Siglec-15 promoter induced by TGF-β1. To exclude the constitutive regulatory effects of ETS-1 and ETS-2 in the region of −453 to −232 nt, the six EBS sites (EBS1 to EBS6) were mutated and the luciferase activities measured in the cells under TGF-β1 stimulation. Intriguingly, the magnitude of the relative luciferase activities remained essentially the same (Figure 5B), which may further consolidate that TGF-β1-induced transcriptional activation of the Siglec-15 promoter was acted through the region of −231 to −22 nt. There were five potential ETS binding sites (EBS1 to EBS5) in the −231 to −22 nt region of the promoter as predicted by JASPAR, Gene-regulation and PROMO programs (Figure 5C). To determine whether ETS-1 and ETS-2 were involved in TGF-β1 mediated transaction of Siglec-15 promoter, pGL4.10-231 was co-transfected with either pcDNA3.1-ETS-1-flag or pcDNA3.1-ETS-2-flag or with siRNAs targeting either endogenous ETS-1 or ETS-2 into HepG2 cells following TGF-β1 treatment. We found that overexpression of ETS-1 or ETS-2 increased the activity of the Siglec-15 promoter luciferase construct; conversely, knockdown of ETS-1 or ETS-2 caused a significant reduction in the luciferase activity (Figure 5D,E). To identify which EBS may be responsible for the TGF-β1-induced transactivation, we generated Siglec-15 promoter constructs with mutations within the individual EBS or deletion of the EBS binding sites. As anticipated, the EBS point mutations (Figure 5F) or deletions (Figure 5G) significantly decreased the promoter activity to varying extent as compared with the control non-mutated or non-deletion construct (pGL4.10-231-Luc) after TGF-β1 stimulation. These results indicate that those EBSs might have a cooperative role together in transactivation of Siglec-15 under TGF-β1 stimulation. To further confirm that TGF-β1-induced transactivation of Siglec-15 worked through the mechanism that involved these EBSs, we again studied the interaction between ETS-1/ETS-2 and these sequences by EMSA. Nuclear extracts from HepG2 cells treated with TGF-β1 were incubated with biotinylated double-stranded oligonucleotides containing the five EBSs. As shown in Figure 5H, the individual oligonucleotide containing the EBS sequence could form a major complex with nuclear proteins from the TGF-β1 treated cells (Lane 2, arrows), which can be completed by an excess unlabeled probe but not by a probe containing a mutation in the EBS site. Addition of an anti-ETS-1 or anti-ETS-2 antibody resulted in the appearance of a supershift (Lane 5 or 6, arrowhead), indicating that ETS-1 or ETS-2 did bind specifically to the EBS site. Notably, TGF-β1-induced transactivation of Siglec-15 through ETS-1 and ETS-2 also translated into the increase of Siglec-15 expression at both mRNA and protein levels (Figure 5I,J). The specificity and efficiency of ETS-1 and ETS-2 overexpression per se was confirmed at both mRNA and protein levels by qRT-PCR (Appendix A) and Western blot analysis (Appendix A).

### 2.6. TGF-β1 Induces Transcriptional Activation of Siglec-15 via MAPK Signaling Pathway

TGF-β1 has been shown to increase ETS-1 expression in different types of cells including cancer cells [9] and regulation of ETS-1 activity at the post-translational levels is mediated mainly by the Ras/C-Raf/MEK/ERK1/2 pathway whose activation leads to phosphorylation of the N-terminus of the ETS-1 protein at threonine-38 and to super-activation of ETS-1 [11]. By using pathway enrichment analysis via the GTBA tool (http://guotosky.vip:13838/GTBA, accessed on 8 May 2022), we found that several biologically important signaling pathways are associated with Siglec-15 gene including the RAS and TGF-β pathway (Figure 6A). Furthermore, a gene set enrichment analysis (GSEA) revealed that Siglec-15 has a high correlation with TGF-β signaling pathways in liver (Figure 6B,C). Therefore, we speculated that TGF-β1-induced Siglec-15 expression may be mediated through activation of Ras/C-Raf/MEK/ERK1/2 pathway that subsequently phosphorylated and activated ETS-1 and ETS-2. To test this hypothesis, HepG2 cells were treated with TGF-β1 and the expression levels of phosphorylated MEK, ERK, ETS-1, ETS-2 were measured by Western blot analysis. Since dephosphorylation of the inhibitory “S259” site on C-Raf kinases plays a key role in C-Raf activation, de-phosphorylated form of C-Raf was also measured to indicate activation of C-Raf. As shown in Figure 6D, treatment with TGF-β1 significantly decreased C-Raf phosphorylation but increased phosphorylation of MEK, ERK1/2, ETS-1 and ETS-2, which was in parallel with an increase of Siglec-15 protein expression. Moreover, the effect was specific as it can be offset by addition of U0126-EtOH, a highly selective inhibitor of MEK1/2 (Figure 6E). Taken together, our data are most consistent with the concept, as pictorially modelled in Figure 6F, that TGF-β1 activates Ras/C-Raf/MEK/ERK1/2 signaling pathway that consequently phosphorylates and activates ETS-1 and ETS-2 which serve as transcription activators to upregulate Siglec-15 expression.

## 3. Discussion

While Siglec-15 has emerged as an attractive immunotherapy target, the exact mechanisms which regulate Siglec-15 expression remain to be elucidated. In the present study, we have demonstrated for the first time that Siglec-15 was constitutively regulated by the transcription factors ETS-1 and ETS-2, and that TGF-β1 upregulated Siglec-15 expression via activation of the Ras/C-Raf/MAPK/ERK1/2 signaling pathway that facilitated phosphorylation and activation of ETS-1 and ETS-2.

The ETS family consists of 28 transcription factors in humans, many of which have been implicated in the development and progression of various cancers. The ETS family contains oncogenic transcription factors in solid tumors. ETSs are ubiquitously expressed transcription factors and there are six putative EBSs identified for ETS-1 and/or ETS-2 in the Siglec-15 promoter region. The EBS motif is generally composed of a purine-rich GGAA/T core that is the binding site for ETS transcription factors [9]. In this study, based on the Siglec-15 promoter activity of EBS mutants, it appeared that the Siglec-15 promoter region spanning nucleotides −453 to −232 nt was crucial for its activity. Using the bioinformatic algorithms to predict ETS-binding sites within this region, we found that there existed six putative EBSs to which either single ETS-1 or both ETS-1 and ETS-2 could bind. However, whether these six ETS-1/ETS-2 binding sites in the Siglec-15 promoter act synergistically or competitively requires further investigation. 

Upregulation of TGF-β is often observed in tumor tissues of most patients with HCC [12]. It has been shown that TGF-β can induce immune escape by upregulation of the expression of PD-1 and cytotoxic T lymphocyte-associated antigen 4 (CTLA-4) on T lymphocytes in HCC to attenuate the cytotoxicity of T lymphocytes for HCC cells both in vitro and in vivo [13]. Studies have shown that Siglec-15 expressed on tumor-associated macrophages (TAMs) may increase TGF-β secretion, thus contributing to its immunosuppression effect [13]. However, whether and how TGF-β1 regulates the expression of Siglec-15 in HCC cells per se remains unexplored. Our experiments in HCC cells validated that TGF-β1 could increase Siglec-15 promoter activity through upregulation of ETS-1 and ETS-2 and consequently activate mRNA transcription and protein expression of Siglec-15. It is well known that genetic and epigenetic alterations can play a crucial role in regulation of cancer development and immune tolerance. Coincidentally, a preliminary analysis indicated that Siglec-15 expression was genetically and epigenetically regulated by copy number variation (CNA) and promoter methylation [14]. GSEA has revealed that Siglec-15 was associated with various signaling pathways such as MAPK, PI3K/Akt, Hippo, p53 and apoptosis, etc. [14]. TGF-β triggers activation of its downstream effectors via canonical and noncanonical pathways. For the classical pathway, TGF-β initially forms a complex with TβRII/TβRI then phosphorylates Smad2/3 transcription factors to facilitate their translocation to the nucleus where they promote the transcription of related genes. Noncanonical pathways involves Wnt/β-catenin, PI3K/Akt, RhoA/ROCK1, MAPK/p38 or MAPK/ERK1/2 that mediate different effect of TGF-β on different tumor cells [15,16,17]. In this study we demonstrated that TGF-β1-induced activation of the Ras/C-Raf/MEK/ERK1/2 signaling pathway increased the phosphorylation and activation of ETS-1 and ETS-2 that bound to the promoter of the Siglec-15 gene to activate the transcription and expression of Siglec-15, which is consistent with previously established observation that ETS-1 and ETS-2 are, respectively, phosphorylated by ERK1/2 at Thr38 and Thr72 [11], and are the major effectors of RAS/MAPK signaling [18]. It is noteworthy that under the physiological condition, the constitutive regulatory effects of ETS-1 and ETS-2 were accomplished mainly through binding to the −453 to −232 nt region of Siglec-15 promoter; however, in the presence of TGF-β1 stimulation, it appeared that ETS-1 or ETS-2 increased Siglec-15 promoter activity via acting on the region of −231 to −22 nt rather than the region of −453 to −232 nt. Since it is well established that post-translational modifications of the ETS family could efficiently affect their DNA binding, protein–protein interactions, transcriptional activation and even subcellular localization [19,20], we speculate that TGF-β1-induced activation of Ras/C-Raf/MEK/ERK1/2 signaling and consequent phosphorylation of ETS-1 and ETS-2 might account for the discrepancy of ETS-1 and ETS-2 binding to the different region of Siglec-15 promoter.

For cancer treatment, tumor heterogeneity is a major challenge since there are different genes that can play different roles via different mechanisms in different cancers. Likewise, Siglec-15 has been reported to have varied prognostic significance in different cancers or even in different subtypes of the same malignancy, reflecting new challenges for studying Siglec-15 [14]. For instance, Siglec-15 may have a role in suppressing tumorigenesis and development in “cold tumors” such as BRCA-luminal A/B; however, Siglec-15 seems to execute an immunosuppressive function in the “hot tumors” like BRCA-basal by regulating immune-related pathways [14]. As for TGF-β, dysregulation of TGF-β signaling is implicated in pathogenesis of some liver diseases including HCC development. Activated TGF-β signaling was shown to associate with an exhausted immune subclass in HCC (approximately 10% of cases) characteristic of depletion of T cells with impaired cytotoxicity [21]. From the therapeutic point of view, the combination of TGF-β inhibition with anti-Siglec-15 antibody may hold great promise for treatment of advanced HCC.

In summary, our results show for the first time that transcription of Siglec-15 was regulated by ETS-1 and ETS-2 in HCC cells. Furthermore, we identify that TGF-β1 signaling induced the expression and activation of ETS-1 and ETS-2, thereby promoting the transcription and expression of Siglec-15. This study is likely to provide a novel insight into how Siglec-15 was transcriptionally regulated in HCC, which might lead to new avenues to enhance immunotherapy efficacy targeting Siglec-15 in this aggressive disease.

## 4. Materials and Methods

### 4.1. Siglec-15 Promoter Luciferase Reporter Constructs

Genomic DNA from HepG2 cells was extracted by a DNeasy Blood and Tissue Kit (QIAGEN, Valencia, CA, USA) and served as a template for polymerase chain reaction (PCR) amplification. The plasmid pGL4.10-1800-Luc with the Siglec-15 promoter driving firefly luciferase was constructed by ligating the PCR-generated full-length Siglec-15 promoter (nucleotides −1800 to +55 nt, relative to the transcription start site) into the XhoI and HindIII (NEB, Ipswich, MA, USA) cleaved sites of the luciferase reporter plasmid pGL4.10-Basic (Promega, Madison, WI, USA). Different Siglec-15 promoter deletion constructs including pGL4.10-1540-Luc (nucleotides −1540 to +55 nt), pGL4.10-1270-Luc (nucleotides −1270 to +55 nt), pGL4.10-942-Luc (nucleotides −942 to +55 nt), p- GL4.10-735-Luc (nucleotides −735 to +55 nt), pGL4.10-453-Luc (nucleotides −453 to +55 nt), pGL4.10-231-Luc (nucleotides −231 to +55 nt) and pGL4.10-21-Luc (nucleotides −21 to +55 nt) were also constructed by cloning the corresponding PCR-generated fragment into the XhoI and HindIII sites of the pGL4.10-Basic plasmid. The primers used in this study for PCR amplification are listed in Appendix A. The pGL4.10-453-Luc plasmid was utilized as a template to generate mutants and deletions of ETS binding sites (EBSs). The resulting plasmids were pGL4.10-453-EBS1-mut-Luc, pGL4.10-453-EBS2-mut-Luc, pGL4.10-453-EBS3-mut-Luc, pGL4.10-453-EBS4-mut-Luc, pGL4.10-453-EBS5-mut-Luc and pGL4.10-453-EBS6-mut-Luc, pGL4.10-231-EBS-1-2-mut-Luc, pGL4.10-231-EBS3-mut-Luc, pGL4.10-231-EBS4-mut-Luc, pGL4.10-231-EBS5-mut-Luc and pGL4.10-231-EBS-1-2-del-Luc, pGL4.10-231-EBS3-del-Luc, pGL4.10-231-EBS4-del-Luc, pGL4.10-231-EBS5-del-Luc. Taq DNA Polymerase High Fidelity (Vazyme, Nanjing, China) was used for PCR amplification, and the amplified PCR products were confirmed by DNA sequencing.

### 4.2. Overexpression Constructs

The plasmid pcDNA3.1-ETS-1-flag or pcDNA3.1-ETS-2-flag was built by insertion of the PCR generated ETS-1 (GenBank Accession No. NM_005238) or ETS-2 (GenBank Accession No. NM_005239) gene into the NheI and HindIII (NEB, Ipswich, MA, USA) sites of pcDNA3.1/Hygro (+). The paired primers used for PCR amplification are listed in Appendix A.

### 4.3. RNA Interference

The small interfering RNA (siRNA) targeting ETS-1 or ETS-2 and a negative control that had no homology with any known human genes were chemically synthesized by GenePharma Company (Shanghai, China). Thirty pmol of ETS-1 siRNA, ETS-2 siRNA (the siRNA sequences are shown in Appendix A) or the negative control was used for transfection by RNAimax (Thermo Fisher Scientific, Waltham, MA, USA) according to the manufacturer’s instructions.

### 4.4. Dual-Luciferase Reporter Assay

The Siglec-15 promoter luciferase reporter constructs were co-transfected with the renilla luciferase expression vector pRL-TK (Promega, Madison, WI, USA) as a normalizer into cells, and the promoter-less vector pGL4.10-Basic served as the negative control. A total of 48 h after transfection cells were lysed and the intracellular luciferase activity in 20 μL of cell lysates was detected by the Dual-Luciferase Reporter Assay System (Promega, Madison, WI, USA) according to the manufacturer’s protocol. Luminescence measurement was performed on an Orion II Microplate illuminometer (Berthold Detection Systems). Each transfection was done in duplicate and the data were presented as the mean ± SD of three independent experiments. 

### 4.5. Cell Culture and Reagents

Human HCC cell lines HepG2, Huh7 and human fetal hepatocyte line LO2 were purchased from American Type Culture Collection (ATCC, Manassas, VA, USA) and authenticated by short tandem repeat profiling analysis at Biowing Applied Biotechnology (Shanghai, China). Cells were cultured in Minimum Essential Medium (MEM) (HyClone: GE Healthcare Life Sciences, Logan, UT, USA) containing 10% (*v*/*v*) fetal bovine serum (FBS) in an incubator with 5% CO_2_ at 37 °C.

### 4.6. Western Blot Analysis

Cells were lysed using ice-cold Western and IP cell lysis buffer (Beyotime Institute of Biotechnology, Jiangsu, China) containing a protease inhibitor cocktail (Med Chem Express, Monmouth Junction, NJ, USA). The protein concentrations of the cell lysates were measured using the Bradford method (Bio-Rad, Hercules, CA, USA). Protein samples were run in 10% sodium dodecyl sulfate polyacrylamide gel electrophoresis (SDS-PAGE) and electrophoretically transferred to a polyvinylidene fluoride (PVDF) membrane (Millipore, Billerica, MA). Membranes were blocked with 5% Skimmed milk powder (BD Biosciences, San Jose, CA, USA)/TBST blocking buffer for 30 min at room temperature and then incubated overnight at 4 °C with individual primary antibodies. Protein blots were incubated separately with a panel of specific antibodies against Siglec-15 (#SAB3500654; Sigma-Aldrich, St. Louis, MO, USA), ETS-1 (#sc-55581; Santa Cruz Biotechnology, Santa Cruz, CA, USA), ETS-2 (#PA5-28053; Invitrogen, Carlsbad, CA, USA) and phospho-ETS2 (Thr72) (#44-1105G; Carlsbad, CA, USA), phospho-ETS-1 (T38) antibody (#ab59179; Abcam, Cambridge, UK), and ETS-1 (#14069), ETS-2 (#66476), GAPDH (#5174), Ras (#3965), C-Raf (#53745), phospho-c-Raf (Ser259) (#9421), MEK1/2 (D1A5) (#13033), phospho-MEK1/2 (Ser221) (#2338), p44/42 MAPK (Erk1/2) (#4695), phospho-p44/42 MAPK (phospho-Erk1/2) (Thr202/Tyr204) (#4370) and the horseradish peroxidase (HRP)-conjugated secondary antibody (#7074), all of which were obtained from Cell Signaling Technology (Danvers, MA, USA). The intensities of the protein signals were quantified by the enhanced BeyoECL Star (Beyotime Biotechnology, Shanghai, China).

### 4.7. RNA Isolation and Real-Time Quantitative Polymerase Chain Reaction

Total RNA was extracted from HepG2 or Huh7 cells using TRIzol Reagent (Invitrogen, Carlsbad, CA, USA) according to the manufacturer’s recommendations. The integrity of the extracted total RNA was evaluated by agarose gel electrophoresis plus ethidium bromide staining. RNA concentration was determined by a NanoDrop-2000 Spectrophotometer system (Thermo Fisher Scientific, Waltham, MA, USA). The reverse transcription reaction was performed with 1 μg of RNA in a final volume of 10 μL by the PrimeScript™ RT reagent Kit (Takara Bio, Shiga, Japan). Quantitative real-time PCR was accomplished with the Mx3000P real-time PCR system (Agilent Technologies, Santa Clara, CA, USA) and the TB Green^®^ Fast qPCR Mix (Takara Bio, Shiga, Japan) following the manufacturer’s recommendations. The primers used for GAPDH, Siglec-15, ETS-1 and ETS-2 amplification are listed in Appendix A. The relative expression level of Siglec-15 was determined by normalization to the internal reference gene GAPDH mRNA expression using the 2^−ΔΔCt^ method.

### 4.8. Electrophoretic Mobility Shift Assay (EMSA)

The 5′-biotin end-labeled oligonucleotides corresponding to the recognition sequences of transcription factor ETS-1 and ETS-2 in the Siglec-15 promoter region were synthesized from Sunya Biotechnology Co., Ltd. (Fuzhou, China) and used as probes. Unlabeled wild-type (cold probes) or mutated oligonucleotides (mutated probes, which included EBS1-mut, EBS2-mut, EBS3-mut, EBS4-mut, EBS5-mut and EBS6-mut) served as competitors. The oligonucleotide sequences are listed in Appendix A. Double-stranded oligonucleotides were made by annealing the complementary single-stranded oligonucleotides with equal amounts (0.1 mg) at 95 °C for 5 min and then gradually cooling down to room temperature. Nuclear proteins from HepG2 cells were extracted using a Nuclear Extraction Kit (Thermo Fisher Scientific, Waltham, MA, USA) according to the manufacturer’s protocol, and the protein concentration of the nuclear extracts was determined by the Bio-Rad DC (detergent compatible) Protein Assay (Bio Rad). An EMSA Gel Shift Kit (Thermo Fisher Scientific, Waltham, MA, USA) was used to examine the interaction between ETS-1 or ETS-2 and Siglec-15 in the nuclear extracts with the probe following the manufacturer’s recommendations. Briefly, 8 μg of nuclear extracts 1 μg of poly (dI-dC) (Sigma-Aldrich, St. Louis, MO, USA) and 0.02 pmol labeled probe were incubated together in a final volume of 10 μL. In the supershift assay, 8 μg of anti-ETS-1 and anti-ETS-2 antibodies was added to the mixture of nuclear extracts and DNA probes. The DNA-protein complexes were incubated at 15 °C for 30 min, run electrophoresis on a 6.0% non-denaturing pre-made polyacrylamide gel and then transferred to a nylon membrane (Amersham Bioscience, Slough, UK) for fixation with a UV crosslinker at 12,000 J for 3 min. The biotin-labeled DNA was visualized by addition of a streptavidin-horseradish peroxidase conjugate and a chemiluminescent substrate.

### 4.9. Flow Cytometry

The cells were trypsinized and resuspended in 100 µL of Fixation/Permeabilization solution (#554714; BD Biosciences, Franklin Lakes, NJ, USA) for broking cell membrane, incubated at room temperature for 20 min, and then washed with 1 mL 1× Perm/Wash Buffer ((#554723; BD Biosciences, Franklin Lakes, NJ, USA) followed by centrifugation at 300 g for 5 min. For the Siglec-15 detection, the cells were incubated with anti-Siglec-15 antibody (1/200 diluted) at room temperature for 1 h then collected and labeled with the FITC secondary antibody (1/1000 diluted) following the manufacturer’s procedure. Cytometry analysis was performed in a FACS CantoII Flow Cytometer (BD Biosciences, San Jose, CA, USA). The data were analyzed by FlowJo v10 (FlowJo LLC, Ashland, OR, USA) and expressed as mean fluorescence intensity (MFI). The antibodies used were as follows: Siglec-15 antibody (#PA5-72765; Invitrogen, Carlsbad, CA, USA) and FITC labeled goat anti-rabbit IgG (H + L) secondary antibody (#F-2765; Invitrogen, Carlsbad, CA, USA. TGF-β1 (#HY-P7118) and SB431542 (#HY-10431) were obtained from Med Chem Express (Monmouth Junction, NJ, USA).

### 4.10. Analysis of Database

To assess the clinical significance of Siglec-15 expression, we first analyzed the Siglec-15 expression level using TCGA data of 369 HCC patients including cancer tissue samples and their adjacent normal tissue samples. The data were analyzed by a statistical programming language, Bioconductor (version 3.2.2; Seattle, WA, USA).

To identify the putative transcription factor binding sites, a computer-based program searching for candidate transcription factor binding site motifs was performed on the Gene regulation professional database by use of Match—1.0 Public (http://gene-regulation.com/cgi-bin/pub/programs/match/bin/match.cgi, accessed on 8 May 2022), JASPAR (http://jaspar.genereg.net/, accessed on 8 May 2022) programs, and PROMO (https://alggen.lsi.upc.es/cgi-bin/promo_v3/promo/promoinit.cgi?dirDB=TF_8.3, accessed on 8 May 2022).

Pathway enrichment analysis and correlation was performed through the GTBA website (http://guotosky.vip:13838/GTBA, accessed on 8 May 2022) using TCGA or the Cancer Cell Line Encyclopedia (CCLE) database.

### 4.11. Statistical Analysis

Flow cytometry data were analyzed using FlowJo v10 software (FlowJo LLC, Ashland, OR, USA). Data are expressed as the mean ± standard deviation (SD) of three separate experiments. Statistical significance was determined with an unpaired, two-tailed Student’s *t*-test and the normality assumption was checked by the Shapiro–Wilk test for each group before performing the Student’s *t*-test. * *p* < 0.05 and ** *p* < 0.01 represent statistical significance. Pathway enrichment analysis and correlation coefficients were analyzed by Pearson’s test. All statistical analyses were carried out using GraphPad Prism 8.0 software (GraphPad Software Inc., La Jolla, CA, USA).

## Figures and Tables

**Figure 1 ijms-24-00792-f001:**
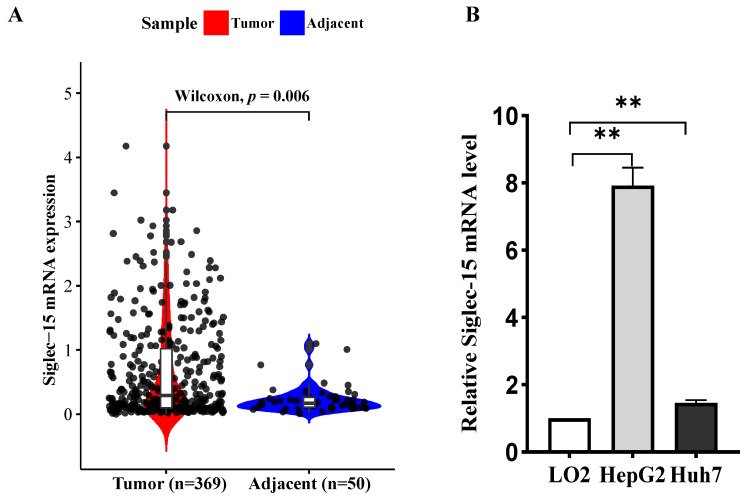
Expression of Siglec-15 in HCC tumors compared to adjacent normal tissues. (**A**) Siglec-15 mRNA expression in HCC and adjacent normal tissues of patients from TCGA database. The data were analyzed by a statistical programming language, Bioconductor, with the test method of Wilcoxon. *p* = 0.006. (**B**,**C**) The mRNA (**B**) and protein (**C**) expression levels of Siglec-15 in the normal embryonic liver cell LO2 and HCC cell lines HepG2 and Huh7 examined by quantitative real-time PCR and Western blot analysis (n = 3; ** *p* < 0.01).

**Figure 2 ijms-24-00792-f002:**
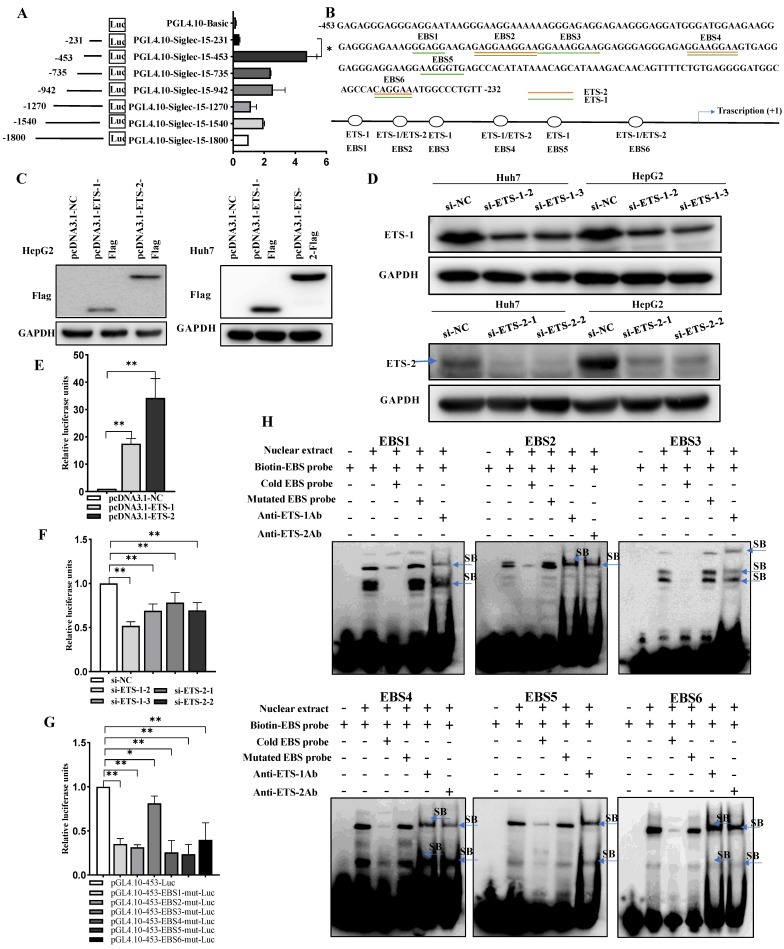
Putative transcription factors and binding sites in the Siglec-15 promoter. (**A**) Activity analysis of the Siglec-15 promoter. HepG2 cells were transfected with each of the Siglec-15 promoter constructs and pRL-TK. The relative luciferase units (RLU) were obtained by comparison with the full-length Siglec-15 promoter construct (pGL4.10-1800-Luc). Each transfection was performed in duplicate and the data are expressed as mean ± SD of three independent experiments. (**B**) Nucleotide sequence of the −453 to −232 region of the Siglec-15 promoter. The six putative binding sites of ETS-1 and ETS-2 were underlined in the sequence and schematically depicted below. (**C**,**D**) Western blot analysis of ETS-1 and ETS-2 expression in HepG2 and Huh7 cells with ETS-1 or ETS-2 overexpressed (**C**) or knocked down (**D**). (**E**,**F**) Luciferase activities in the cells with ETS-1 or ETS-2 overexpressed (**E**) or knocked down (**F**). (**G**) Effect of EBS mutations on Siglec-15 pGL4.10-453 promoter activity. EBS mutants (0.25 μg) together with 10 ng of pRL-TK were co-transfected into HepG2 cells. The RLU were obtained by comparison with the wild-type pGL4.10-453-Luc. (**H**) Electrophoretic mobility shift assays with specific antibodies against ETS-1 and ETS-2. Arrowheads indicate the shift band (SB). The biotin-labeled probe was incubated in the absence (Lane 1) or presence (Lane 2) of nuclear extracts from HepG2 cells; unlabeled cold probe (Lane 3) and mutated probe (Lane 4) were used as competitor at a concentration of 100-fold molar excess to biotin-labeled probe. Super-shift assays were carried out with 8 μg specific antibody raised against ETS-1 or ETS-2 (Lane 5 or Lane 6). * *p* < 0.05; ** *p* < 0.01. NC stands for negative control.

**Figure 3 ijms-24-00792-f003:**
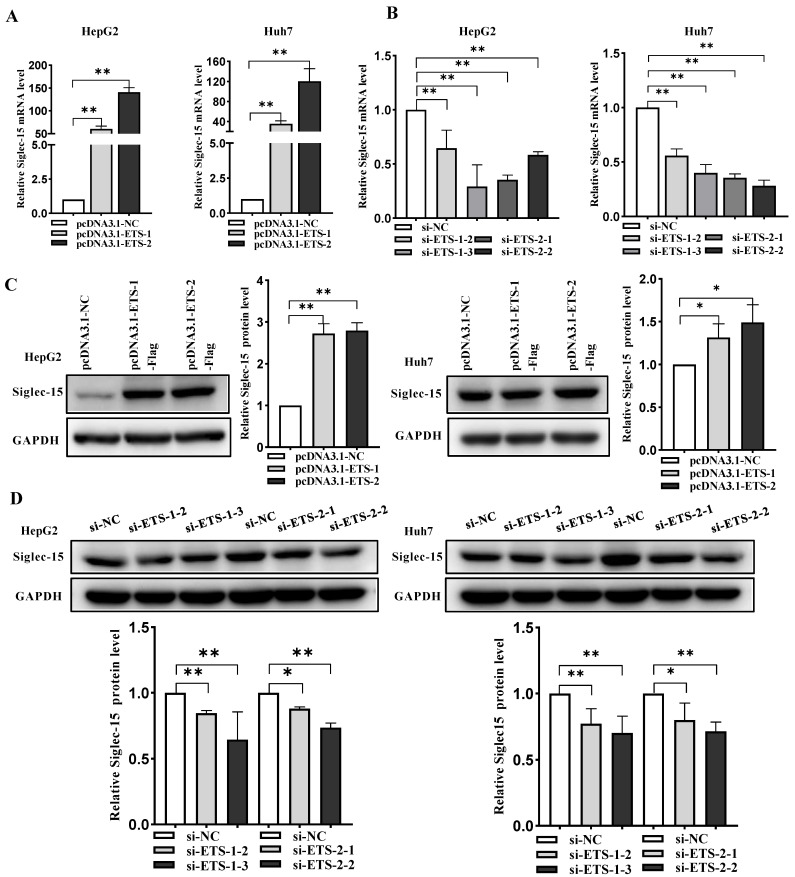
ETS-1 and ETS-2 enhances transcription and protein expression of Siglec-15 in HepG2 and Huh7 cells. (**A**,**B**) Effect of ETS-1 or ETS-2 overexpression (**A**) or knockdown (**B**) on Siglec-15 expression in the transfected HepG2 or Huh7 cells by quantitative real-time PCR. (**C**,**D**) Effect of ETS-1 or ETS-2 overexpression (**C**) or knockdown (**D**) on Siglec-15 expression in the transfected HepG2 or Huh7 cells by Western blot analysis. * *p* < 0.05; ** *p* < 0.01. NC stands for negative control.

**Figure 4 ijms-24-00792-f004:**
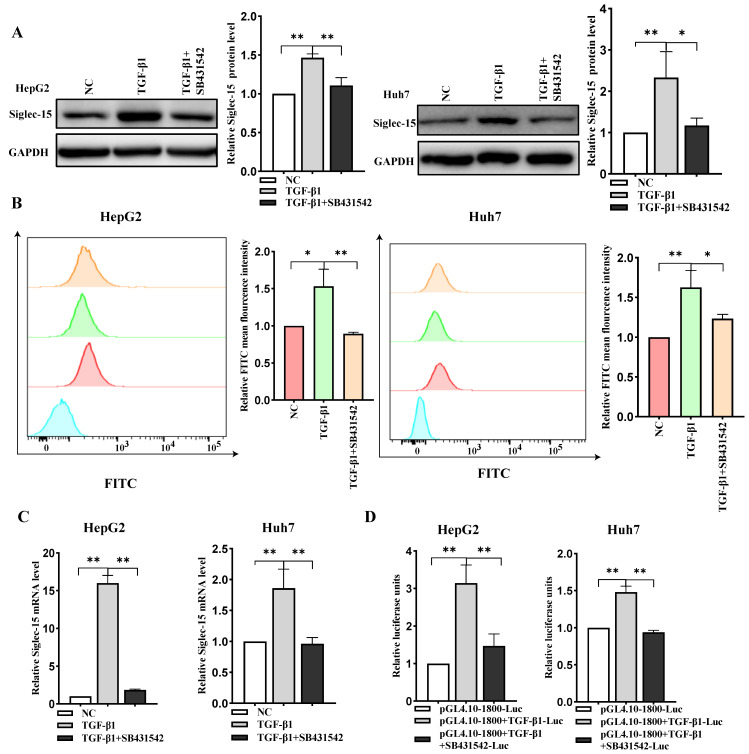
TGF-β1 upregulated Siglec-15 expression in HCC cells. (**A**) Western blot analysis of Siglec-15 protein levels in the cells treated with TGF-β1 (50 ng/mL) for 24 h in the absence or presence of 8 mM SB431542 for 12 h. (**B**) Flow cytometric analysis of Siglec-15 expression in the HepG2 and Huh7 cells with the same treatment as specified above. * *p* < 0.05; ** *p* < 0.01. NC stands for untreated control. (**C**) Quantitative real-time PCR analysis of Siglec-15 mRNA expression in the same treated cells. (**D**) Dual-luciferase reporter assay for TGF-β1 sensitivity in HCC cells. Cells were transfected with the luciferase reporter vector and then treated with or without 50 ng/mL TGF-β1 for 24 h in the absence or presence of 8 mM SB431542 for 12 h.

**Figure 5 ijms-24-00792-f005:**
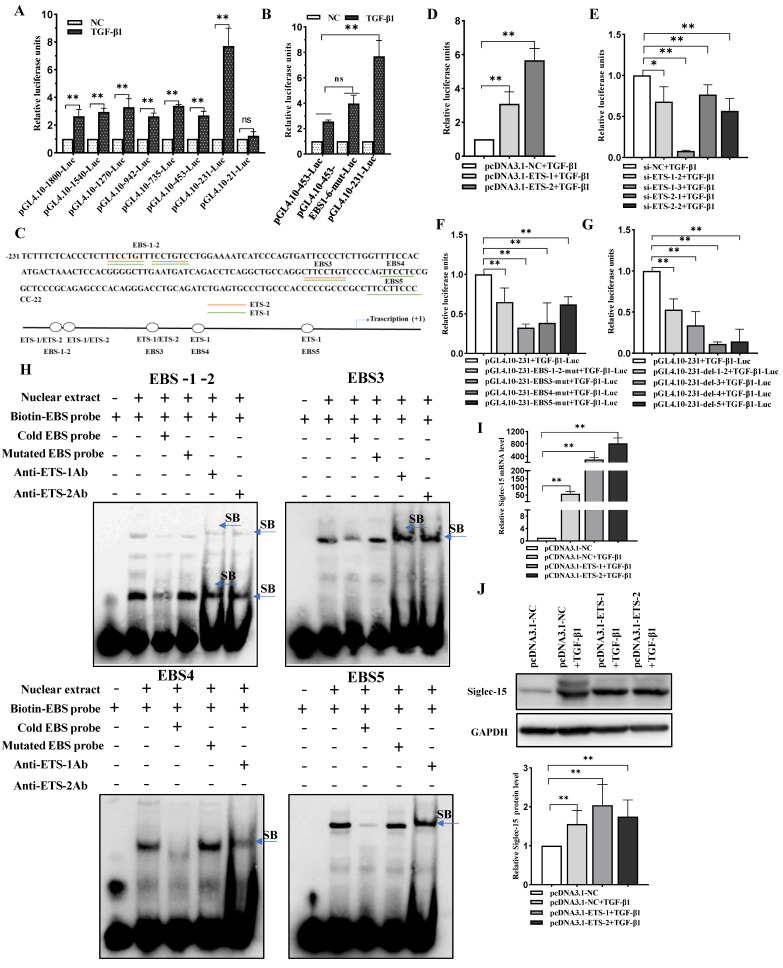
TGF-β1 upregulated the expression of Siglec-15 in HepG2 cells by facilitating the binding of ETS-1 and ETS-2 to EBSs in the Siglec-15 promoter. (**A**) Activity analysis of the Siglec-15 promoter in HepG2 cells treated with TGF-β1 (50 ng/mL) for 24 h and then transfected with each of the Siglec-15 promoter constructs and pRL-TK. (**B**) Luciferase activity in HepG2 cells treated with TGF-β1 (50 ng/mL) for 24 h and then transfected with pGL4.10-231-Luc, pGL4.10-453-Luc or the mutant plasmid with all the six EBSs mutated. (**C**) Nucleotide sequence of the −231 to −22 nt region of the Siglec-15 promoter. The five putative transcription factor binding sites are underlined in the sequence and schematically depicted below. (**D**,**E**) Effect of TGF-β1 on Siglec-15 promoter activity in HepG2 cells with ETS-1 or ETS-2 overexpression or knockdown. pcDNA3.1-ETS-1, pcDNA3.1-ETS-2, si-ETS-1, or si-ETS-2 was co-transfected with pGL4.10-231-Luc into HepG2 cells treated with TGF-β1 (50 ng/mL) for 24 h, then luciferase activities were measured 48 h after transfection. The RLU were obtained by comparison with empty vector pcDNA3.1/Hygro (+) or siRNA negative control (NC), respectively. (**F**,**G**) Effect of EBS mutation or deletion on TGF-β1-induced Siglec-15 promoter activity. The variants of EBS mutations or deletions (0.25 μg) were co-transfected with 10 ng of pRL-TK into HepG2 cells treated with TGF-β1 (50 ng/mL) for 24 h. The RLU was calculated by normalizing with the wild-type pGL4.10-231-Luc. (**H**) Electrophoretic mobility shift assays with specific antibodies against ETS-1 and ETS-2. Arrowheads indicate the shift band (SB). The biotin-labeled probe was incubated in the absence (Lane 1) or presence (Lane 2) of nuclear extracts from HepG2 cells treated with TGF-β1 (50 ng/mL) for 24 h. Unlabeled cold probe (Lane 3) or mutated probe (Lane 4) was used as a competitor at a concentration of 100-fold molar excess to biotin-labeled probe. Super-shift assays were carried out with 8 μg specific antibody raised against ETS-1 or ETS-2 (Lane 5 or Lane 6). (**I**,**J**) Effect of ETS-1, ETS-2 and TGF-β1 on Siglec-15 mRNA and protein expression in HepG2 cells determined by quantitative RT-PCR (**I**) and Western blot analysis (**J**). * *p* < 0.05; ** *p* < 0.01. ns stands for not significant and NC for untreated control.

**Figure 6 ijms-24-00792-f006:**
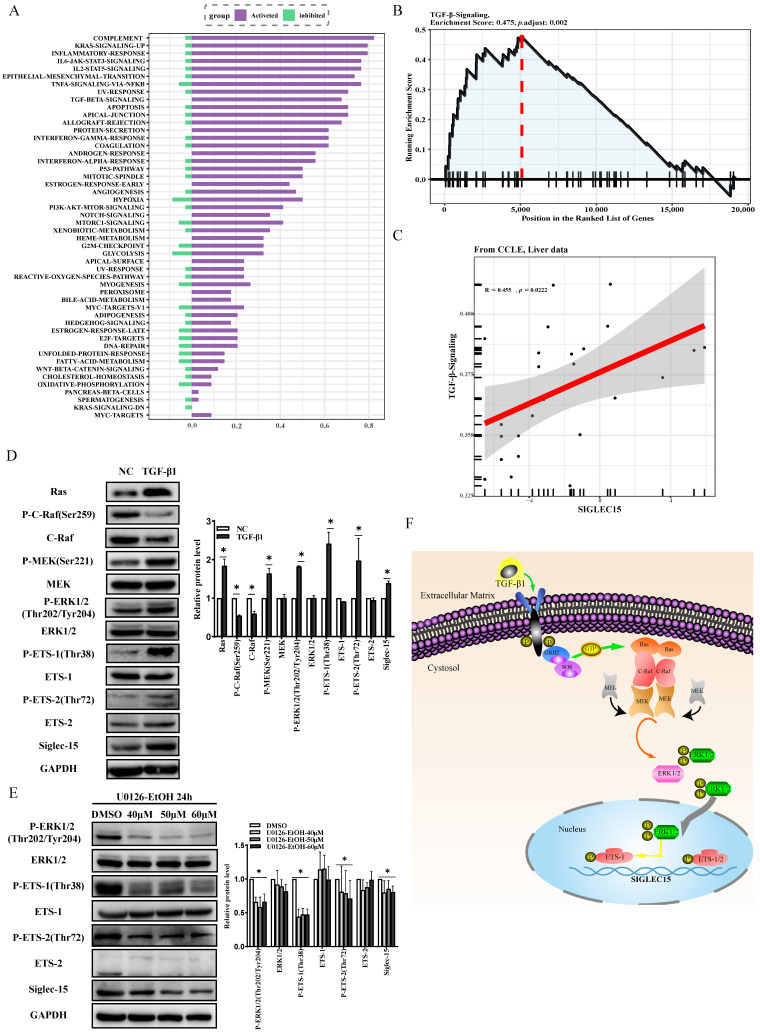
TGF-β1 induces transcriptional activation of Siglec-15 via MAPK signaling pathway. (**A**) Enrichment analysis of signaling pathways associated with Siglec-15 using The Cancer Genome Atlas Liver Hepatocellular Carcinoma (TCGA-LIHC) dataset through GTBA platform. (**B**,**C**) The signaling pathway enrichment analysis (**B**) and correlation analysis (**C**) between Siglec-15 and TGF-β based on GTBA analysis using CCLE liver data source. (**D**) The expression levels of Ras/C-Raf/MEK/ERK1/2 signaling molecules, ETS-1/ETS-2 and their phosphorylation as well as Siglec-15 in HepG2 cells by Western blot analysis. (**E**) Effect of U0126-EtOH, a highly selective MEK1/2 inhibitor, on the expression of ERK1/2, ETS-1/2 and their phosphorylated forms as well as well Siglec-15 in HepG2 cells by Western blot analysis. (**F**) Schematic model for the mechanism by which TGF-β1 activates Ras/C-Raf/MEK/ERK1/2 signaling to phosphorylate and activate ETS-1 and ETS-2 leading to upregulation of Siglec-15 expression. * *p* < 0.05. NC stands for untreated control.

## Data Availability

Data will be provided upon reasonable request.

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
