# Peer review of "Transcriptional Regulation of Siglec-15 by ETS-1 and ETS-2 in Hepatocellular Carcinoma Cells"

_ijms, 2023, doi:10.3390/ijms24010792_

Round 1

Reviewer 1 Report

General comments:

The Paper “Transcriptional regulation of Siglec-15 by ETS-1 and ETS-2 in hepatocellular carcinoma cells” by Sheng et al. shows that the expression of the immunosuppressive protein Siglec15 is transcriptionally regulated via TGFbeta/MAPK cascade and ETS transcription factors. Their work is sound, however there are some details that require further explanation and clarification. As well, in some experiments controls are not shown.

Specific comments:

·         Figure 2D: knockdown efficiency of siETS-1-2 and siETS-1-3 not convincing. Authors should comment on the efficiency of the two siRNAs. How do the authors explain that siRNAs against ETS1 also knockdown ETS2? This may also explain the decrease of luciferase activity in Figure 2F, which is probably only due to ETS2 inhibition. Please clarify.

·         Figure 3 A-B: ETS1 and ETS2 mRNA levels as controls for specific overexpression/specific knockdown are missing. Authors need to provide ETS1/2 mRNA measurements to prove specificity and efficiency.

·         Figure 3C-D: WB images of ETS1/ETS2 detection should be shown to prove overexpression and knockdown efficiency and specificity.

·         Figure 4B: the colors of the histogram and the barplot are different, which is misleading for the reader. For example, NC should have the same color in the histogram and in the barplot (e.g.green). Please change.

·         Figure 5 is in principle a repetition of Figure 2. How do the authors explain the fact, that (as shown in Figure 2) ETS1/2 promote Siglec regulation at a region between -230 and -450 and exclude the region of 22-232 while in Figure 5 they show that ETS1/2 bind to this region? This is contradictive and needs explaining.

·         Furthermore, in Figure 5I and J control levels of ETS1/2 mRNA and protein should be shown as proof for overexpression.

·         Figure 6 is blurry.

Reviewer 2 Report

Transcriptional Regulation of Siglec-15 by ETS-1 and ETS-2 in Hepatocellular Carcinoma Cells

Manuscript ID: ijms-2020716

Summary:

In this study by Sheng K and et. al., they have identified the phosphorylated ETS1 and its closely related ETS2 (55% overall amino acid identity, 70% similarity) participate along the TGF-b1 signaling pathway to modulate Siglec-15 overexpression in two HCC cell lines (HepG2 and Huh7).

Areas of Strength: The authors have demonstrated that ETS-1 and ETS-2 play an equal role in Siglec-15 expression in HCC. TGF-b1 was also earmarked to be essential for Siglec-15 protein expression using a TGF-b1 inhibitor. The authors later identified individual EBS regions as potential siRNA therapeutic target along the Siglec-15 promoter region to be critical for TGF-b1 induced transcriptional activation.

Areas of Weakness: Although there is statistical significance across the measured changes, the relative changes to Siglec-15 protein level due to ETS-1/-2 knockdown or phosphorylation and TGF-b1 are minimal. Some of the figures are not readable due to poor resolution. Also, there are some data inconsistencies to be addressed below. Lastly, an ithenicate analysis shows certain sentences in the introduction need to be further rephrased to minimize plagiarism.

1. Whether the EBS mutants are deletion or point mutations is unclear in the main text without referring to the supplementary.

2. siRNA knockdown using the mutated EBSs was concluded to be statistically significant in two different HCC cell lines using the Student’s t test. It remains unclear if the authors have conducted a normality check before performing a Student’s t test analysis instead of Mann-Whitney.

3. The same pcDNA3.1-NC is used as reference in Figure 2E and 5D. However, there is data inconsistency when comparing the delta change in luciferase protein level in Figure 2E (+20 to 40, ETS-1/2) and Figure 5D. In Figure 5D, the relative increase due to ETS-1/2 + TGF-b1 is significantly reduced. Can the authors explain the loss of luciferase activity when comparing Figures 2E and 5D? 

4. In Figure 2H: It is convincing that unmutated EBS1, EBS2, EBS3, EBS4, EBS5, EBS6 on the same Siglec15 promoter can individually disrupt the formation of nascent DNA-transcription factor complex formation in vitro. In contrary, in Figure 2G using HepG2 cells, the loss in luciferase activity is not greater than it should be. Perhaps, the authors can provide a possible explanation.

5. In Figure 3D: To correct “Si-“ to “si-“.

6. Figure 4B: To apply the same color scheme used to the different treatment groups (NC, TGF-b1, TGF-b1 + SB431542).

7. Figure 4B: y-axis mislabeling “indensity” instead of “intensity”.

8. Figure 4B: to provide a higher resolution as the axis label is unclear (in HepG7 and Huh7) or cropped (in Huh7).

9. Figure 4D: Missing in the Figure 4. Is this Figure 4B?

10. Figure 5B and 5E: In Figure 5B, the relative luciferase unit change from 4 to 3 is insignificant. But in Figure 5E, the relative luciferase unit change for siRNA knockdown from 1.0 to 0.8 is regarded as statistically significant. Perhaps, showing scatter points in the same bar plots will be useful for identifying outliers.

11. Figure 5C: The annotated EBS regions do not match Figure 2B despite using the same naming convention e.g., EBS3.

12. Table S1: Correct Muted to “Mutated”

Round 2

Reviewer 1 Report

The labelling of the siRNAs in Figure 2D is still the same in V2 of the manuscript. Please correct as specified in your answer to the reviewer's comments.

Author Response

As corrected, in the upper panel of Fig. 2D two siRNAs against ETS-1, namely si-ETS-1-2 and si-ETS-1-3, were applied to either Huh7 or HepG2 cells for knocking down ETS-1 expression; Likewise, both of the two specific siRNAs targeting ETS-2, i.e. si-ETS-2-1 and siETS-2-2, were used to knock down ETS-2 expression in either of the two cell lines (Bottom panel of Fig. 2D).

Reviewer 2 Report

The authors have made the necessary major edits. However, Figures 6A, B and F are still in low-resolution. The scientific observations are now statistically sound and is appropriate for highlighting the Siglec-15 promoter's role in driving TGF-b1 induced transcriptional activation in cancer.

Author Response

We are very pleased to know that our previous revision was made to the reviewer’s full satisfaction. As recommended, a higher resolution of Fig. 6A has now substituted the old one.